# E-Governance and Political Modernization: An Empirical Study Based on Asia from 2003 to 2014

**Shouzhi Xia**

Graduate Institution of National Policy and Public Affairs, National Chung Hsing University, Taichung 402, Taiwan; g104091026@mail.nchu.edu.tw

**Abstract:** This study aims to analyze whether E-governance matters for political modernization in Asia. According to the literature review, E-gcovernance can be operated by three elements: open data, online service, and E-participation. Political modernization can also be divided into three elements: the government's transparency, the offline political participation, and the level of liberty. Analyzing second-hand data from several databases, this study draws such conclusions. Firstly, the development of E-governance will lead to the improvement of political modernization in Asia. Specifically, open data have a positive impact on the government's transparency. E-participation has a positive impact on the offline political participation and the level of liberty. Secondly, it is difficult to confirm which aspect of E-governance has the greatest impact on political modernization, as open data and E-participation have an impact on different aspects of political modernization. Based on the result, Asian countries should emphasize the importance of E-governance so that political modernization in this region could be improved continuously.

**Keywords:** E-governance; political modernization; Asia

## 1. Introduction

How to build an efficient, reliable, transparent, and democratic government is an enduring topic. With the aim of figuring out this conundrum, some scholars raised the theory of New Public Management (NPM) to replace the traditional public administration (Dunleavy and Hood 1994; Kaboolian 1998); some others raised another way, which is called New Public Service (NPS), a movement built on democratic citizenship, community and civil society, and organizational humanism and discourse theory (Denhardt and Denhardt 2000). Both NPM and NPS are different from the traditional public administration, so they can be seen as innovative governance. The rapid development of the Internet, information and communication technologies (ICTs) creates a juncture to put the spirit of these theories into practice. Specifically, E-government benefits from the Internet, and ICTs can accelerate government reform. As time goes by, more and more public problems need to be resolved by the E-government system or E-governance (Heeks 2001; Prabhu 2013).

E-governance can be roughly divided into two aspects: (1) open government, which means government offers the official data and online service for citizens; and (2) E-participation, which means governments set up some online approaches for citizens to participate in public discussions and the process of policy-making (Tolbert and Mossberger 2006). In the past one or two decades, many scholars have concentrated on the aspect of open government, aiming to analyze whether the E-government system can improve government's working efficiency. According to a considerable number of empirical studies, E-governance has a positive impact on this field (Torres et al. 2005; Potnis 2010). However, there is a lack of studies attempting to analyze the influence of E-participation on political development. Even though some scholars aim to involve this topic, their vision is limited to the local or domestic layer. Meanwhile, some studies have announced that E-governance could not

improve political modernization significantly (Torres et al. 2005; Mazzarella 2006; Noesselt 2014). However, we cannot draw such a conclusion that E-governance has no or limited impact on political modernization, as many studies have obtained different results; i.e., E-governance's political influence is evident (Zheng and Zheng 2014; Kudo 2010; Iqbal and Seo 2008; Tan et al. 2005; Lewis and Litai 2003).

Asia is a typical region affected by E-governance. For instance, Iqbal and Seo (2008) showed that E-governance was a good way to reduce corruption in South Korea. Kudo (2010) held the view that E-governance played a role in the public sector's reform of Japan, especially in relation to public accountability. Some scholars pointed out that the advanced E-government system in Singapore has shown its function, as public trust in the government has increased significantly (Tan et al. 2005). E-governance development in China is also very eye-catching. The Chinese government, at both central and local levels, has done many meaningful actions to improve government efficiency and transparency.[1] Meanwhile, to adapt to social development, the government has started setting up official accounts on Weibo, Wechat as well as other social media (Zheng and Zheng 2014).[2] By the end of June 2016, more than 170 million net users had ever used government's official accounts (China Internet Network Information Center 2016), which is a remarkable outcome.

However, there have been few empirical studies illustrating E-governance's impact on political modernization using cross-national analysis in Asia. With no doubt, political modernization today is still an important topic, so whether E-governance has a significant influence on political modernization in Asia is a worthwhile question. Thus, this paper aims to resolve this conundrum through collecting and analyzing second-hand data from UN databases, Transparency International (TI), and Varieties of Democracy (V-Dem). There are three main questions. First, to what extent does E-governance matter for political modernization in Asia? Second, which factor of E-governance has the greatest impact on political modernization? As mentioned above, E-governance can be divided into different aspects, but what aspect has the most significant influence on political modernization is yet to be resolved. This paper tries to clarify this. Third, what should Asian countries do in response to these results?

## 2. Literature Review

### 2.1. Political Modernization and Its Components

Political modernization is a relatively complex concept. After World War II, or even since the period of Enlightenment, many countries around the world have searched for approaches to reach political modernization, and many of these movements can be seen as the process of democracy to many degrees. In short, political modernization can be regarded as the development of politics in many aspects. Meanwhile, some scholars have pointed out that political modernization is based on democratic institutions (Gould 1990). For instance, Warren (1996) stated that deliberative democracy is a good way to handle authority and to make political decisions. Estlund (2009) held the view that how political decisions are made is one of the key indicators for evaluating the legitimacy of an authority, stating:

> democracy is a way of giving every (adult) person an equal chance to influence the outcome
> of the political decision, even though democracy has no particular tendency to produce
> good decisions (Estlund 2009, p. 8).

---

[1]  For example, Beijing's open government data contain more than 400 datasets, including tourism, education, transportation, land use zoning and medical treatment. People living in Beijing can access government website and gain information freely. Besides "open data", people today can also use E-government to participate in public affairs. On the Chinese government's Ministry of Environment Protection website, people can provide their opinions on government document drafts, which might be received by government (United Nations 2016).

[2]  The development of Internet and social media in China is significant. By the end of June 2016, the total number of Chinese net users had reached 710 million, and the number of social media users reached more than 550 million (China Internet Network Information Center 2016). Thus, the Chinese government has emphasized the importance of social media by setting up official social media accounts to serve people and offer official data and materials for interested citizens.

Basically, political modernization means the structure of authority has changed significantly, which also indicates the development of rationalization and legitimacy of authority. Specifically, political modernization, according to the opinion raised by Max Weber, means the source of authority changing from tradition and Charisma to legitimacy.[3] In pre-modern society, authority had two primary sources: (1) traditional way, which means political leaders inherit authority through the ties of blood; and (2) Charisma way, which means political leaders gain authority relying on their glamour, talent, or other characteristics. These two types of authority would lead to centralism to a high degree. In Western countries, however, thanks to Enlightenment and political reforms, the source of authority has changed fundamentally. To be more specific, citizens' votes are the essential resource of political leaders' authority today. In other words, rationalization or legitimacy of authority widely exists in Western countries. While in other places, e.g., many countries in East and Southeast Asia, Central America and Africa, the development of political modernization is slower than in Western countries.

Political modernization is behind the process of economic and social modernization in non-Western countries. For instance, in East and Southeast Asian countries, e.g., South Korea, Singapore, Taiwan, the rapid rise in economic status undoubtedly influenced political development. However, the progress of economy and society might also lead to several problems in developing countries, especially corruption. As Huntington (2011) said,

> Impressionistic evidence suggests that its (corruption) extent correlates reasonably well with rapid social and economic modernization [ . . . ] The differences in the level of corruption which may exist between the modernized and political developed societies of Atlantic world and those of Latin American, Africa, and Asia in lager part reflect their differences in political modernization and political development (pp. 253–54).

In the context of such a background, government's transparency is necessary. Moreover, in some scholars' opinions, the increasing level of governments' transparency will lead to the development of political modernization (Relly and Sabharwal 2009). In the past, due to the low level of governments' transparency, ordinary people had no or minimal access to official information, which leads to serious corruption, especially in those undemocratic or semi-democratic nations with rapid economic growth (Sung 2004). Thus, building a transparent government is necessary. The transparent government means that all political processes, from policy-making to policy implementation, could be supervised by ordinary people. Even though improving the level of government's transparency is widely accepted by almost every country, there are still many examples of corruption and black boxes in Asia. How to strengthen government's transparency is becoming a hot topic in this region. Some scholars pointed out that the development of technologies might be a functional way of accelerating the improvement in a government's transparency. For instance, E-governance, with a rapid diffusion in Asia, has been proved to have a positive impact on government's transparency (Zheng and Zheng 2014; Kudo 2010; Iqbal and Seo 2008; Tan et al. 2005).

Besides, offline political participation is workable when facing the serious corruption (Huntington 2011). However, as many scholars have pointed out, political participation is affected by several factors, e.g., social structure, national history, tradition, etc. (Nie et al. 1969; Verba et al. 1987; Pye and Pye 2009). Thus, in some political-developing countries or regions, including many Asian countries, political engagement is increasing slowly, even though some scholars have pointed out that civil society was strengthened in Asia over the past two or three decades. They cited many cases to support their opinion, including the 1986 mass protest for president Marco's ouster in the Philippines and the highly mobilized civil society in South Korea that compelled President Chun Doo Hwan to accept the demand of opposition in 1987 (Alagappa 2004). However, in spite of these countries

---

[3]   The authority from tradition and Charisma to legitimacy means that government is the product of man, not of nature or of God, and that a modern society must have a determinate human source of final authority, obedience to whose positive law takes precedence over other obligations (Huntington 2011).

achieving democratization after social movement and political reform, the civic engagement in many other countries in this region is still at a low level. In the recent decade, the rapid development of the Internet and other information tools, however, apparently promoted the growth of offline political participation (Dalton 2013). For example, in the Chinese mainland, E-governance's influence could be witnessed. In the past, people could hardly take part in political issues because of the lack of suitable approaches, but with the diffusion of social media and the completed E-government system, people today have a way to participate in public affairs online, which can also arouse people's awareness of the offline political participation (Zhang 2006; Zhang and Chan 2013; Zheng and Zheng 2014). Many other similar cases can be found in Asia, but there seems to be a lack of research attempting to study the E-governance's influence on offline political participation by analyzing cross-nations' data.

Overall, government's transparency and offline political participation are essential aspects of political modernization. Meanwhile, the liberal political environment is also irreplaceable for political modernization. In practice, if the political environment is not liberal or even autocratic, both citizens' participation and the transparent government would not appear. In other words, the high level of liberty is a fundamental element that leads to the development of political modernization. Thus, I point to the view that political modernization involves three aspects, namely: governments' transparency, offline political participation, and the level of liberty.

## 2.2. E-Governance and Its Functions

As mentioned above, how to build an efficient, transparent, and democratic government is an enduring topic. However, the level of public trust in the government is falling, in both democratic and undemocratic countries. The government has to find a right way to improve the efficiency and transparency. In this context, E-government came into being. According to the definition proposed by the UN and American Society for Public Administration (ASPA), E-government is utilizing the Internet for delivering government information and services to citizens (Torres et al. 2005). Organization for Economic Cooperation and Development (OECD) gave another illustration, which is using ICTs and particularly the Internet as a tool to achieve better government (Torres et al. 2005). Based on these definitions, we find that the essential purpose of E-government is improving the efficiency and transparency of government through offering official information and online service (Janssen and Estevez 2013; Torres et al. 2005; Chadwick and May 2003). E-government's influence is obvious based on many empirical studies. For example, Shim and Eom (2008) found that E-government had a positive impact on anti-corruption. In the past, people almost had no access to official data, including financial budget and the process of policy-making. This situation means that public employees could escape from supervision to some extent. The development of E-government, however, creates a new way for citizens to get close to relevant materials including financial budget and policy-making process. As a result, the situation of corruption will fall.

As time goes by, the concept of "governance" has replaced some traditional concepts like government, administration, and management. Many scholars and institutions had raised different definitions of governance. For example, United Nations Educational, Scientific, and Cultural Organization (UNESCO) defined governance as structures and processes that are designed to ensure accountability, transparency, responsiveness, rule of law, stability, equity and inclusiveness, empowerment, and broad-based participation (UNESCO 2016). Rhodes (1996) held the view that governance means self-organizing, inter-organizational networks that complement markets and hierarchies as governing structures for authoritatively allocating resources and exercising control and coordination. There are still many other definitions. Summing up these ideas, we can conclude that the government is no longer the only decision-maker: corporations, citizens, and non-government organizations (NGO) have played an irreplaceable role in public affairs. In this context, the concept of E-governance had also been put forward. Traditional E-government could not meet the requirement of "good governance", even though citizens could get official data and receive an online service through the Internet. Traditional online systems lack participatory function so that ordinary people could not take

part in public affairs online. However, E-government's function has deepened today, so many scholars have pointed out that E-governance is much more accurate than E-government (e.g., Torres et al. 2005).

> E-governance includes E-government plus key issues of governance such as online engagement of stakeholders in the process of shaping, debating and implementing public policies (p. 278).

In the beginning, the primary purpose of E-government was offering online data so that ordinary people could become more knowledgeable about government and public affairs. Then, many online services were supplied by the government, which aim to facilitate people's daily life. With the help of E-government system, people can accomplish many things online, such as paying tax, submitting documents and so on, which had to be done offline in the past. These two functions, open data, and online service, have much common ground to some extents. Both of them can improve the efficiency and transparency of the government, so according to some scholars' opinion, these two functions together can be regarded as open government (Tolbert and Mossberger 2006). Political participation was the new functional development of E-governance. Online participation or E-participation means citizens can express their opinions, take part in the public issues discussion and even monitor government and officers through online platforms. In the past, these political rights were hard to put into practice, as there was a lack of suitable approaches. E-governance, without a doubt, offers citizens an appropriate and workable approach to achieve their political rights. As a result, this function is also called E-democracy (Tolbert and Mossberger 2006). Overall, the functions of E-governance can be divided into three aspects, which are: open data, online service, and E-participation.

### 2.3. The Political Influence of E-Governance in Asia

Many pieces of research have focused on the E-governance's influence on political development. Many of these studies obtained positive results, in other words, the development and innovation of E-governance can bring a higher level of political modernization (Ciborra 2005; Madon 2008). For instance, Chadwick and May (2003) pointed out that E-governance enshrines some important norms and practices of E-democracy, even though the potential for linking E-democracy in civil society with E-governance at the level of the local and national state is far from straightforward to some extent. Many scholars have also pointed out that E-governance's influence could be witnessed in Asia. For example, the Chinese government paid more attention to the transparency of the government and achieved significant results in recent years. To some degrees, the rapid development of E-governance, especially the aspect of open government, is one of the most important elements accelerating the improvement of the Chinese government's transparency (Jun et al. 2014). In summary, E-governance's influence on political modernization, according to the previous studies, could be divided into three aspects, which are governments' transparency, offline political participation and the level of liberty.

Some scholars have stated that a government's transparency can be improved through some specific channels, such as proactive dissemination by the government, releasing of requested materials by the government, public meetings, and leaking from whistle-blowers (Piotrowski and Ryzin 2007). E-governance can provide these channels. First, people can get official data and the online service that they need from online systems. Second, with the development of the Internet and ICTs, online discussion and E-participation have been popularized from country to country, which brings an opportunity for citizens' political participation. Taking the EU, for example, this region enjoys advanced E-government systems, which is a good condition for increasing transparency and customer-oriented service that aid good governance in this area (Torres et al. 2005).

In addition to the EU, the influence of E-governance on the government's transparency in Asia is also evident. For example, E-governance in South Korea plays a significant role in anti-corruption. Corruption can be widely found in both democratic and undemocratic countries. The only distinction lies in its degree. South Korea, a relatively developed country in Asia, had experienced a rapid

economic growth that has been racked by severe corruption. How to resolve this issue is one of the primary tasks faced by the South Korean government. Some claimed that the anti-corruption movement could make a big difference through E-governance. The online systems in South Korea have an impact on anti-corruption such as the Online Procedure ENhancement (OPEN) system for civil applications of Seoul Metropolitan Government (SMG), and the Government e-Procurement System (GePS) will be analyzed and this will generate policy implications for reducing corruption (Iqbal and Seo 2008, p. 53).

In short, E-governance's development offers a functional way to enhance transparency. Singapore is another typical representative in the region. As some scholars pointed out, the advanced E-government system in Singapore has shown its function because the government is more transparent and the public trust in the government has also increased (Tan et al. 2005). The reason why citizens in Singapore show a higher level of trust in the government is that they could get more and more information, enjoy the online service, and take part in public discussions through E-government system. Based on the statement above, this paper raises the first group of hypotheses.

*H1a.* *The development of open data will lead to a higher level of the government's transparency in Asia.*

*H1b.* *The development of online service will lead to a higher level of government's transparency in Asia.*

*H1c.* *The development of E-participation will lead to a higher level of government's transparency in Asia.*

Political participation is an important indicator to evaluate the level of countries' democracy. In the past, political participation was hard to achieve because of the lack of suitable approaches. The E-government system offers a new approach for citizens to take part in public issues. In the online platform, citizens can express their opinions about public affairs and even monitor the process of policy-making. In practice, scholars also concentrate on the E-governance's effect on people's offline political participation. For instance, taking the 100 largest cities of America for example, Scott (2006) found that some cities' websites were useful and could improve citizens' involvement, but in other cities, it was hard to find the same effect, in other words, official websites in these cities had no or minimal effect on civil engagement. As time goes by, especially with the development of the Internet (e.g., Web 2.0), the political influence of E-governance has been more and more significant. For example, some scholars focused on biodiversity governance, they analyzed 2000 networks in Finland, Greece, Poland, and the UK, and found that citizens in those countries widely engaged in policy-making processes related to the environment protection (Paloniemi et al. 2015).

In the recent decade, Asian countries, especially East Asian countries, have emphasized the importance of building an excellent E-government system. The Chinese government, for example, aims to improve its efficiency and transparency through the E-government system. With the development of the E-government, other fields are also affected, including offline political participation. Jiang and Xu (2009) stated that citizens' political participation might generate unintended consequences of incremental reform of China's local governance and political institutions based on the development of E-governance. He et al. (2017) found that more and more Chinese ordinary people today take part in the policy-making process of environment protection, expressing their opinions or even attending offline political activities. We can understand that E-governance, especially the aspect of open data and E-participation, affects ordinary people's attitudes toward offline political participation based on the statement above. Thus, this paper raises the second group of hypotheses.

*H2a.* *The development of open data will lead to a higher level of offline political participation in Asia.*

*H2b.* *The development of E-participation will lead to a higher level of offline political participation in Asia.*

A few studies focused on the E-governance's influence on the level of liberty. However, the impact of E-governance on citizens' liberty seems evident. First, citizens are more liberal to get official data when the E-government system has been built completely. In the past, this type of

liberty was not enjoyed by citizens, as they can hardly gain the information they needed from the government, even if they ask for it through a legal approach. People today can get what they need from a government's website or database. Meanwhile, they can call for extra data liberally through online platforms. Second, citizens are more free to enjoy government's services, as they could choose online services offered by E-government systems or undertake this business offline. Before the diffusion of E-governance, this situation did not exist or was even beyond people's imagination. In other words, citizens had to handle business through a series of complex and fussy processes and face, perhaps, ill-mannered government employees. Nowadays, things have changed, much business can be done online with an offline-type outcome through online systems. Third, citizens today enjoy the liberty to participate in the public discussion through online platforms, which could not have been imagined two or three decades ago. Overall, the development of E-governance exactly brings a higher level of liberty to citizens. As Relly and Sabharwal (2009) emphasized that one of the key elements that E-governance brings to our societies is a more liberal lifestyle. They thought this lifestyle could also promote so-called good governance. Thus, this paper raises the third group of hypotheses.

*H3a:* *The development of open data will lead to a higher level of liberty in Asia.*

*H3b:* *The development of online service will lead to higher level of liberty in Asia.*

*H3c:* *The development of E-participation will lead to a higher level of liberty in Asia.*

### 3. Methodology

*3.1. Data Source*

Three databases are used by this study: UN database, TI[4] and V-Dem[5]. First, independent variables and control variables (Economic growth, education, and cultural tradition) are sourced from the UN database. Second, data of governments' transparency is sourced from TI where studies for Corruption Perceptions Index (CPI) have been taken since 1995. Third, relevant data of political participation and the level of liberty can be obtained from the V-Dem database.

Two things should be illustrated here. First, the UN E-governance Survey began in 2001, and in total nine surveys had been held so far. However, there was a clack of data of E-participation in the first UN E-government Survey, and the data included in V-Dem ended in 2014. As a result, in total, seven surveys can be used in this study. Second, to ensure that all of the relevant data mentioned above exist, this study removes the countries lacking the whole or a part of the data. Eventually, in total, 30 countries are included in this study.[6]

*3.2. Variables' Operationalization*

*Open data* Based on UN E-government Survey, Telecommunication Infrastructure Index (TII), evaluating the status of telecommunication infrastructure, can be regarded as the essential condition for open data's improvement. In other words, if the telecommunication infrastructure is not developed well, people can hardly get the information or data from the online platform. In contrast, if the TII index of a country is very high the telecommunication basis in this nation is perfect and open data

---

[4]   Transparency International was found in 1993, aiming to improve governments' transparency around the world. It began to report CPI (Corruption Perceptions Index) since 1995.

[5]   V-Dem (Varieties of Democracy) is one of the largest-ever social science data collection efforts with a database containing over 16 million data points. By April 2017, the dataset will cover 177 countries from 1900 to 2016 with annual updates to follow.

[6]   Countries list: Afghanistan, Azerbaijan, Bangladesh, Bhutan, Cambodia, China, Cyprus, Georgia, India, Indonesia, Israel, Japan, Jordan, Kazakhstan, Lao, Malaysia, Mongolia, Myanmar, Nepal, Pakistan, Philippines, South Korea, Sri Lanka, Syrian Arab Republic, Tajikistan, Thailand, Timor-leste, Turkey, Turkmenistan, and Vet Nam.

will also be developed remarkably well. Specifically, TII has been included in the UN E-government Survey since 2001. This index is between 0 (the worst) to 1 (the best).

*Online service* Based on the UN E-government Survey, the Online Service Index (OSI), evaluating the level of online service in the world, can be used in this study. If a country has a high score of OSI, this means the level of online service is relatively high. In contrast, if the score is at a low level, it means a high-quality online service is lacking. Thus, the score of OSI can represent the aspect of online service. Specifically, this index is also between 0 (the worst) to 1 (the best).

*E-participation* The factor of E-participation aims to evaluate the situation of citizens' online political participation, which is also one of the key parts in the UN E-government Survey. In the Survey, the E-Participation Index (EPI) is raised to represent the situation of people's online political participation in different countries, so the score of EPI can be used in this research. This index is also from 0 (the worst) to 1 (the best).

*The government's transparency* The government's transparency represents the level of transparency in different countries. The TI database started to evaluate this situation by setting up the Corruption Perceptions Index (CPI) since 1995. This is the annual report about CPI in the TI database. Thus, this paper regards the score of CPI as the situation of the government's transparency. CPI was from 0 (highly corrupt) to 10 (very clean) before 2012. After that time, it was changed from 0 (highly corrupt) to 100 (very clean). In order to gain the same scale of CPI from 2003 to 2014, this paper standardizes and uses the scale from 0 (highly corrupt) to 1 (very clean).

*The offline political participation* The factor of offline political participation aims to test the situation of ordinary people's political engagement in practice. Civil Society Participation Index (CSPI) sourcing from the V-Dem database can be regarded as the representative of offline political participation. This index aims to provide a measure of a robust civil society, understood as one that enjoys autonomy from the state and in which citizens freely and actively pursue their political and civic goals. The questions used to measure CSPI do not include anything about Internet or E-participation, so these two indexes, EPI and CSPI, are different from each other. In Summary, the score of CSPI can represent the offline political participation in this study, and it is from 0 (lowest) to 1 (highest).

*The level of liberty* The level of liberty can be roughly divided into civil liberty and political liberty. Two indexes together can represent this variable. The first one is Civil Liberty Index (CLI), which is measured by some questions related to the level of absence of physical violence committed by government agents and the level of absence of constraints of private liberties and political liberties by the government. The second one is the Political Liberty Index (PLI). Among the set of civil liberties, these liberal rights are the most relevant for political competition and accountability. The index is based on indicators that reflect government repression and that are not directly referring to elections. The level of liberty can be regarded as CLI plus PLI, which is from 0 (lowest) to 1 (highest). Thus, the range of the level of liberty is from 0 (lowest) to 2 (highest).

Besides, this part will introduce the situation of control variables. First, according to many previous studies, economic growth can be seen as an essential element having an impact on political modernization (Lipset 1959; Arat 1988; Bernstein 1971; Przeworski and Limongi 1997). Based on the literature review, this study regards the annual GDP growth as the representative of economic growth. This data can be sourced from the UN database. Second, education's influence on political modernization has also been proved by a considerable number of empirical studies (Coleman 2015; Han 2000; Brown 2007). In this paper, the data of public expenditure on education as the percentage of total government expenditure (%) (range from 0 to 1) sourced from the UN database can be seen as the representative of education's development in Asia. Third, different cultural traditions have a distinct impact on political modernization (Inglehart and Baker 2000; Pye and Pye 2009). Based on the situation in Asia, cultural traditions can be roughly divided into four types: Confucian, Muslim, Buddhism, and other traditions.

## 4. Results

### 4.1. Descriptive Analysis

When we concentrate on three aspects of E-governance, it is obvious that there is a big gap from country to country, especially focusing on the aspect of E-participation. This phenomenon indicates that Asian countries have performed differently in E-governance. Meanwhile, there is the least distinction between different countries in online service, considering its score, Asian countries have shown a poor performance in this aspect. Besides, when we consider the situation of political modernization in this region, the differences are also very evident. The score of government's transparency is from 0.150 to 0.733, and the mean score is 0.322, which shows that the level of government's transparency in Asia is low. Offline political participation is from 0.170 to 0.899, which also indicates significant gaps among different countries. The biggest gap among Asian countries is the level of liberty. In the end, when considering these control variables, it is easy to find that economic growth and government expenditure on education are very distinct from country to country in Asia (see Table 1).

**Table 1.** Data description.

|  | N | Min | Max | Mean | S.D. |
|---|---|---|---|---|---|
| Open data | 210 | 0.004 | 0.730 | 0.213 | 0.228 |
| Online service | 210 | 0.030 | 0.425 | 0.192 | 0.119 |
| E-participation | 210 | 0.023 | 0.872 | 0.296 | 0.239 |
| The government's transparency | 210 | 0.150 | 0.733 | 0.322 | 0.172 |
| Offline political participation | 210 | 0.170 | 0.899 | 0.620 | 0.229 |
| The level of liberty | 210 | 0.171 | 1.876 | 1.179 | 0.586 |
| Economy | 210 | 0.018 | 0.097 | 0.057 | 0.509 |
| Education | 210 | 0.011 | 0.051 | 0.031 | 0.478 |
| Cultural tradition | 210 | 1.000 | 4.000 | 2.900 | 1.062 |

### 4.2. Regression

First, from the Table 2, we can gain the following information. According to the relevant score ($p = 0.000 < 0.05$), this model can be accepted, and the high adj. R-squared, 0.752, indicates that the independent variables account for a substantial amount of government's transparency. In detail, open data ($\beta = 0.785$, $p < 0.001$) has a significant effect on government's transparency. In theory, if this element grows by one unit, then the level of government's transparency will increase by 0.785 units. However, online service ($\beta = 0.253$, $p > 0.05$) and E-participation ($\beta = -0.104$, $p > 0.05$) have no significant impact on government's transparency. As for control variables, education ($\beta = 0.042$, $p < 0.05$) has a significant effect on government's transparency, even though the degree of its influence is not very high.

**Table 2.** Determinants model of government's transparency.

|  | β | S.E. | VIF | *p* |
|---|---|---|---|---|
| Open data | 0.785 | 0.095 | 1.302 | 0.000 |
| Online service | 0.253 | 0.870 | 2.118 | 0.091 |
| E-participation | −0.104 | 0.637 | 2.447 | 0.211 |
| Economy | 0.122 | 0.426 | 4.302 | 0.051 |
| Education | 0.042 | 0.238 | 3.021 | 0.022 |
| Culture tradition (Confucian = 0) |  |  |  |  |
| Muslim | −0.017 | 0.224 | 3.875 | 0.081 |
| Buddhism | 0.158 | 0.201 | 2.906 | 0.112 |
| Others | 0.039 | 0.131 | 2.977 | 0.247 |
| Adj. R$^2$ = 0.752; *p* = 0.000; N = 210 | | | | |

[a] Dependent variable: the government's transparency.

Second, from the Table 3, we can get the following findings. According to relevant score ($p = 0.003 < 0.05$), this model can be accepted, but the low adj. R-squared, 0.268, reveals that the explicability of this model is not very high. To be more specific, open data ($\beta = 0.350$, $p > 0.05$) has no significant impact on offline political participation. In other words, people's willingness to undertake political participation will not be impacted evidently, even if the government has offered enough official data. However, E-participation ($\beta = 0.632$, $p < 0.05$) has a significant effect on people's offline political participation. Besides, these control variables have no significant impact on people's offline political participation.

**Table 3.** Determinants model of offline political participation.

|  | β | S.E. | VIF | *p* |
|---|---|---|---|---|
| Open data | 0.350 | 0.780 | 1.128 | 0.134 |
| E-participation | 0.632 | 0.224 | 1.650 | 0.042 |
| Economy | 0.032 | 0.353 | 3.920 | 0.121 |
| Education | 0.042 | 0.238 | 4.115 | 0.102 |
| Culture tradition (Confucian = 0) | | | | |
| Muslim | 0.406 | 0.284 | 3.875 | 0.288 |
| Buddhism | 0.596 | 0.186 | 2.906 | 0.083 |
| Others | 0.514 | 0.169 | 2.977 | 0.174 |
| Adj. $R^2$ = 0.268; $p$ = 0.003; N = 210 | | | | |

[a] Dependent variable: offline political participation.

Third, from the Table 4, we can find the determinants model of liberty. According to the relevant score ($p = 0.000 < 0.05$) this model also can be accepted, but its explicability is at a low level (adj. R-squared is 0.225). To be more specific, open data ($\beta = 0.053$, $p > 0.05$) and online service ($\beta = 0.328$, $p > 0.05$) have on significant impact on the level of liberty. However, E-participation ($\beta = 0.624$, $p < 0.05$) has a significant impact on the level of liberty. As for control variables, education ($\beta = 0.102$, $p < 0.05$) is also the only one that has a significant effect on the level of liberty.

**Table 4.** Determinants model of liberty.

|  | β | S.E. | VIF | *p* |
|---|---|---|---|---|
| Open data | 0.053 | 0.423 | 1.302 | 0.123 |
| Online service | 0.328 | 0.305 | 2.118 | 0.311 |
| E-participation | 0.624 | 0.577 | 2.447 | 0.013 |
| Economy | 0.093 | 0.294 | 4.302 | 0.133 |
| Education | 0.102 | 0.301 | 3.021 | 0.034 |
| Culture tradition (Confucian = 0) | | | | |
| Muslim | −0.422 | 0.224 | 3.875 | 0.081 |
| Buddhism | 0.355 | 0.201 | 2.906 | 0.112 |
| Others | −0.519 | 0.131 | 2.977 | 0.247 |
| Adj. $R^2$ = 0.225; $p$ = 0.000; N = 210 | | | | |

[a] Dependent variable: the level of liberty.

## 5. Discussion

First, many possible reasons can illustrate open data's impact on governments' transparency in Asia. The most important one is that ordinary people can get more and more information from the government and they can also supervise government employees relying on open data. For instance, Beijing's open government contains more than 400 datasets, including tourism, education, transportation, land-use zoning and medical treatment. People living in Beijing can surf these websites, gain relative information liberally, and monitor public employees of these departments (United Nations 2016; Lollar 2006). In addition, the South Korean government's transparency has

also experienced obvious growth with the help of E-governance. In order to tackle corruption, many online platforms have been built by the central or local South Korea governments, such as the Online Procedure ENhancement (OPEN), the Government e-Procurement System (GePS), and so on. All of these online platforms are built for offering information and avoiding black boxes, which brought an ideal result (Iqbal and Seo 2008).

Second, the most important reason, which can be used to illustrate E-participation's influence on the offline political participation, is that online political participation has cultivated ordinary people's awareness of attending public affairs. Without a doubt, online political participation is simpler, more time-saving and money-saving than taking part in public affairs offline. Thus, many people are willing to attend public discussions, voting and other types of political participation through an E-government system. As time goes by, the willingness of people's political participation will be stronger, and if there is no feedback from online platform or citizens are unsatisfied with government's actions, they are likely to attend offline political activities. For example, in Sweden, people are called to participate in public affairs through the Internet, such as online discussions and online voting. According to the data, citizens' willingness of political participation has clearly been growing, especially offline political participation (Phang and Kankanhalli 2008).

Third, it is not difficult to understand the influence of E-participation on the level of liberty. In general, the level of liberty in Asia is not at a high level, which is affected by many elements. Thus, citizens cannot fully enjoy political rights, including freedom of speech, demonstration and so on. With the help of E-governance, especially E-participation, people can practice or even extend their political rights in many aspects. For instance, people can enjoy a higher level of freedom of speech through online platforms. China is a typical case. Even though Chinese citizens still cannot express all of their opinions, especially negative discourses about the central government and political highest-level persons, people today can voice their views to the local government and its officers. Meanwhile, a considerable number of views could receive serious feedback (Yang 2009). In addition, the improvement of E-participation means people have more appropriate approaches to play a role in public affairs, which means a huge change in Asia, as many people living in this region had no or inadequate access to political affairs in the past (Chen et al. 2006). In this context, the level of liberty in Asia is increasing widely.

## 6. Conclusions

First, according to the statement above, we can conclude that E-governance has a positive impact on political modernization in Asia. To begin with, open data can promote the growth of governments' transparency, but two other elements, online service, and E-participation have no significant effect on this field. This finding is similar to many previous empirical studies. For example, Relly and Sabharwal (2009) stated that telecommunication infrastructure (TI) influenced the perceptions of government transparency in a positive and significant way. In addition, E-participation can be regarded as the explanatory variable for the improvement of offline political participation and the level of liberty. In contrast, online service and open data have no significant influence on these two fields in a theoretical context. This finding is different from previous empirical studies that underestimated E-participation's function (Saglie and Vabo 2009; Goldfinch et al. 2009).

Second, it is difficult to confirm which element has the greatest influence on political modernization, as open data and E-participation have an impact on the different aspects of political modernization. At such background, it is better to discuss this agent separately. Specifically, when considering the aspect of governments' transparency, there is no doubt that open data should be emphasized preferentially. However, when the offline political participation and the level of liberty are considered, the importance of E-participation is greater than the other two aspects of E-governance.

Third, regarding what Asian countries should do to respond to the result, I think there might be two approaches. The first one is building online system and offering official data continually. Based on the statement above, open data's impact on government's transparency is very evident so Asian

countries should emphasize the importance of open data and regard this element as the effective way of building a transparent government. As Bertot et al. (2010) pointed out, the combination of E-government, Web-enabled technologies, transparency policy initiatives and citizen desire for open and transparent government are fomenting a new age of opportunity that has the potential to build a more transparent and reliable government. The second one is emphasizing the importance of E-participation. The enhancement of E-participation means that government liberalizes the restrictions on citizens' political participation, so ordinary people have more and more opportunities to play a role in public affairs, both online and offline. Without a doubt, it is the right way to improve the level of political participation and liberty which characterizes political modernization in Asia.

In conclusion, this study has proved the impact of E-governance on political modernization in Asia. However, some shortcomings of this paper should be promoted in the future, such as the means of operating variables, the volume of the sample, and so on.

**Conflicts of Interest:** The authors declare no conflict of interest.

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
