# Peer review of "E-Governance and Political Modernization: An Empirical Study Based on Asia from 2003 to 2014"

_admsci, doi:10.3390/admsci7030025_

Round 1

Reviewer 1 Report

Wordings in abstract are too affirmative, especially before "adequate justification is provided", such as proofs (proves?)..make it clear that...

Introduction is fairly written.

Lit review is good, yet some logical connectives can better be enhanced

Some issues in headings and sub-headings issues..

Methodology part is good

Results can better be elaborated, esp some themes can be extracted for further in-depth discussion like open data..transparency...

Conclusion is fairly done. 

Author Response

First, as the statement in the beginning of the article, the reason why focusing on East and Southeast Asia is that there are many political-developing countries in this place with a rapid development in E-governance based on UN E-government Survey. So does E-governance matter for political modernization or development can be made clear when considering the situation in this region. It should be recognized the number of counties in this study is limited, but considering the topic of this study, I don’t think there is a essential barrier for this study to gain a reasonable result. Second, I think the literature review has included the relative things mentioned by the review.   

Reviewer 2 Report

I  would like to start out by commenting on the number of cases in the  research design: the authors present three cross-sectional linear  regression models consisting each of 13 countries  in South East and East Asia. While it is not clear what a minimum  cut-off point for such analyses would be, there is consensus that the  more observations in the analysis (also in relation to the number of  countries), the better. 13 strikes me as too few.

            I think that the number of observations could be increased by  either working with panel data and using several measurement times, i.e.  country years, or to include other countries in the analysis. While the  authors refer to several existing contributions  in the literature that focus on South East and East Asia, I don't see  why the points they are trying to make wouldn't also apply for other  countries in the world. After all, the possible effects of E-governance  on modernization would not just apply to authoritarian  countries in South East and East Asia but it would also be testable and  of great interest to highly democratic and digitally native states in  other world regions, such as like Finland or Estonia. In such a design  using an increased number of countries, one  could then still test for whether South East and East Asia is different  from the rest.

            However, my most serious concern with the draft under  consideration is that no reference is being given to the vast empirical  literature that exists for explaining country-level variation on the  study's three dependent variables, i.e. transparency, participation,  and liberty. I would ask the authors to go back to the drawing board  and very seriously consider the vast number of work for each one of  their dependent variables, then pick the relevant sources as candidates  to be replicated, and finally present regression  models augmented with their three independent variables, i.e. open  data, online services, and E-participation. Currently, there are no  control variables whatsoever which is not acceptable at all and a bit of  an affront to all the existing statistical work  in the field of democratization.

Author Response

(The authors gave the same response as above.)
